# Retinitis Pigmentosa Associated with EYS Gene Mutations: Disease Severity Staging and Central Retina Atrophy

**DOI:** 10.3390/diagnostics13050850

**Published:** 2023-02-23

**Authors:** Giorgio Placidi, Paolo Enrico Maltese, Maria Cristina Savastano, Elena D’Agostino, Valentina Cestrone, Matteo Bertelli, Pietro Chiurazzi, Martina Maceroni, Angelo Maria Minnella, Lucia Ziccardi, Vincenzo Parisi, Stanislao Rizzo, Benedetto Falsini

**Affiliations:** 1Ophthalmology Unit, Fondazione Policlinico Universitario Agostino Gemelli IRCCS, Largo Gemelli 8, 00168 Rome, Italy; 2Ophthalmology Unit, Catholic University of the Sacred Heart, Largo Francesco Vito 1, 00168 Rome, Italy; 3MAGI’S LAB, 38068 Rovereto, Italy; 4MAGI EUREGIO, 39100 Bolzano, Italy; 5MAGISNAT, Atlanta Tech Park, 107 Technology Parkway, Peachtree Corners, GA 30092, USA; 6Medical Genetics, Fondazione Policlinico Universitario Agostino Gemelli IRCCS, Largo Gemelli 8, 00168 Rome, Italy; 7Genomic Medicine, Catholic University of the Sacred Heart, Largo Francesco Vito 1, 00168 Rome, Italy; 8IRCCS-Fondazione Bietti, 00198 Rome, Italy; 9Consiglio Nazionale delle Ricerche, Istituto di Neuroscienze, 56127 Pisa, Italy

**Keywords:** retinal degeneration, EYS gene, disease staging, multimodal imaging, OCT, subretinal illumination, electroretinography

## Abstract

Background. Eyes shut homolog (EYS) gene mutations are estimated to affect at least 5% of patients with autosomal recessive retinitis pigmentosa. Since there is no mammalian model of human EYS disease, it is important to investigate its age-related changes and the degree of central retinal impairment. Methods. A cohort of EYS patients was studied. They underwent full ophthalmic examination as well as assessment of retinal function and structure, by full-field and focal electroretinograms (ERGs) and spectral domain optical coherence tomography (OCT), respectively. The disease severity stage was determined by the RP stage scoring system (RP-SSS). Central retina atrophy (CRA) was estimated from the automatically calculated area of the sub-retinal pigment epithelium (RPE) illumination (SRI). Results. The RP-SSS was positively correlated with age, showing an advanced severity score (≥8) at an age of 45 and a disease duration of 15 years. The RP-SSS was positively correlated with the CRA area. LogMAR visual acuity and ellipsoid zone width, but not ERG, were correlated with CRA. Conclusions. In EYS-related disease, the RP-SSS showed advanced severity at a relative early age and was correlated with the central area of the RPE/photoreceptor atrophy. These correlations may be relevant in view of therapeutic interventions aimed at rescuing rods and cones in EYS-retinopathy.

## 1. Introduction

Pathogenic variants in the eyes shut homolog (EYS) gene are estimated to affect at least 5% of patients with autosomal recessive retinitis pigmentosa (arRP) [1]. The gene is defective in several arRP populations worldwide [2,3,4,5]. EYS encodes a large extracellular protein [6] that in Drosophila promotes the formation of epithelial lumina, a selective space inside the rhabdomeres which isolates individual photoreceptor cells and is useful for their development [7]. In humans, the function of EYS is not yet fully understood. However, the protein product is thought to play a role in stabilizing ciliary axonemes in rods and cones and it is involved in the maintenance of photoreceptor cells [8]. Since there is no mammalian model of human EYS disease, it is important to investigate its age-related retinal changes in both structure and function, as well as the degree of central retina involvement.

Recently, Iftikhar et al. [9] developed a simple and easily applicable classification of disease severity in RP patients, with a score based on best corrected visual acuity (BCVA), Goldmann visual field diameter and ellipsoid zone (EZ) width. This RP stage score system (RP-SSS) appeared to be readily applicable to different subtypes of RP. In particular, our group [10] recently applied the RP-SSS to the clinical and morphological features of USH2A related retinal degeneration. The USH2A severity score was reliably correlated with patient age as well as with several morphologic and functional parameters of retinal disease [10].

In a previous study by Mcguigan et al. [11], EYS patients showed typically RP features. Although a residual foveal cone function could still be detected into the fourth, fifth or sixth decade of life, the outer nuclear layer (ONL) thickness and EZ width progressively decreased. Determining the degree of retinal atrophy involving retinal pigment epithelium (RPE)/photoreceptors in the macular region may be particularly relevant either for visual prognosis or for the effects of potential treatments rescuing photoreceptors. An estimate of the area of central macular atrophy (CRA) can be obtained by the automatic determination of the area of sub-RPE illumination (SRI). The SRI identifies bright areas of increased light transmission beneath the RPE, indicating RPE/outer retina atrophy, averaged over a circular area of 5 mm around the fovea by the automated spectral domain optical coherence tomography (SD-OCT) software. In age-related macular degeneration, the SRI was used to measure the area of RPE and outer retinal atrophy (RORA), established from an international consensus by Guymer et al. [12].

The aim of the present study was to determine the disease severity stage in a cohort of ArRP patients carrying pathogenic variants in the EYS gene and to estimate, in these patients, the extent of CRA by using the automated SRI measurements.

## 2. Materials and Methods

This clinical study was performed in compliance with the ICH Guidelines for Good Clinical Practice, adhered to the tenets of the Declaration of Helsinki (1991) and was approved by the Ethics Committee/Institutional Review Board of the Catholic University of Rome, Italy (protocol #8383/15).

After a detailed explanation regarding the study procedures, written informed consents for clinical and molecular analyses were obtained from all the adult subjects or relatives when the patient was a minor.

All the reported clinical data were retrospectively re-evaluated.

### 2.1. Subjects

We enrolled 17 patients (8 male, 9 female; mean age: 50.5 years.; SD: ±14.7) affected by RP due to variants in the EYS gene and followed at the Center for Inherited Retinal Degenerations of Fondazione Policlinico A. Gemelli, IRCCS. All patients were evaluated between November 2013 and December 2021 and met the following inclusion criteria: (1) clinical and genetic diagnosis of EYS-related RP; (2) good cooperation in psychophysical testing; (3) dioptric media clean enough to perform Spectral Domain-Optical Coherence Tomography (SD-OCT). Exclusion criteria were the presence of: (1) concomitant ocular (e.g., amblyopia, glaucoma) and systemic diseases; (2) poor cooperation in psychophysical testing; (3) severe ocular media opacities. EYS-related RP diagnosis was confirmed through a genetic test and, whenever possible, a segregation analysis on available family members was performed. Seven patients were homozygous and the remaining eight were compound heterozygous for EYS variants. Molecular genetic data are reported in Table 1. The pathogenic variants in the EYS gene were detected using the Next Generation Sequencing (NGS) technology. NGS was performed on a MiSeq personal sequencer (Illumina, San Diego, CA), following the molecular and bioinformatic strategy that we previously published [13,14]. Multiplex ligation-dependent probe amplification (MLPA) (www.mrc-holland.com, (accessed on 1 December 2017) was also performed in one patient using the Beckman Coulter CEQ 8000 sequencer, and revealed the presence of a deletion in heterozygosity (c. (2135 _2204) _ (2351_2469) del) resulting in the loss of the exons 14 and 15. All variants identified were evaluated according to American College of Medical Genetics and Genomics (ACMG) guidelines [15] with the help of the VarSome online tool (https://varsome.com/, (accessed on 24 May 2022) [16]. Molecular genetic data for each patient are detailed in Table 1.

### 2.2. Clinical Assessment and Functional Evaluation

All patients underwent a full ophthalmologic examination including detailed family history, anterior segment biomicroscopy, intraocular pressure measurement, BCVA measured with ETDRS charts, Goldmann visual field using the V/4e target, SD-OCT with measurement of the EZ extension, scotopic and photopic full-field electroretinogram (ERG) recordings and direct and indirect ophthalmoscopy. All patients had a typical RP phenotype. The data collected allowed the determination of the disease severity stage of all enrolled patients according to the cumulative score (CS) and grade indicated by Iftikhar et al [9]. Sub-groups of patients underwent a more detailed comprehensive electro-functional study. A total of 12 out of 17 patients performed 30 Hz submicrovolt flicker ERG, (Retimax Advanced Plus, CSO, Scandicci, Italy)) with the assessment of response variability and signal-to-noise ratio (S/N), as already described in a previous article by Falsini et al. [10].

Macular cone-mediated focal ERG (FERG) was recorded in 7 out of 17 patients using a published technique [17,18,19,20]. Briefly, FERGs were recorded monocularly in response to a flickering uniform red field stimulus superimposed on an equiluminant steady adapting background. Off-line discrete Fourier analysis quantified the peak-to-peak amplitude of the response first harmonic at 41 Hz.

### 2.3. Morphological Analysis Using Retinal Advanced Multimodal Imaging

All patients underwent advanced retinal analysis by SD-OCT using Zeiss Cirrus 5000-HD-OCT Angioplex, sw version 10.0, (CarlZeiss, Meditec, Inc., Dublin, CA, USA). In two patients, the advanced imaging was not possible due to unstable fixation. A High-Definition 5-Line Raster and a macular map (6 × 6 mm Macular Cube 512 × 128) were acquired. In order to obtain the most reliable measurements, two independent operators (M.C.S. and M.M.) measured the residual EZ extension on OCT horizontal scans using a caliper, as previously described [21]. The agreement rate between the two independent experts was equal to 89% (95% confidence interval = 79–98%).

The EZ extension was determined by the retinal points where the temporal and nasal EZ borders met the RPE becoming indistinguishable.

The Advanced RPE post-processing analysis was used to automatically determine areas of sub-RPE illumination (SRI, measured in mm^2^) for increased light penetration through atrophic OR, RPE and choriocapillaris, by means of the sub-RPE algorithm. An automated SD-OCT software allowed detection of RPE atrophy in a 5 mm circular area around the fovea. The SRI was used to measure the RPE and outer retinal atrophy (RORA), as established in an international consensus by Guymer et al. [12].

RORA corresponds to a region of signal hyper-transmission into the choroid resulting from the interruption of the RPE and OR and can be classified as complete and incomplete. If the signal hyper-transmission into the choroid does not exceed an area of 250 microns, RORA is defined as incomplete (iRORA). RORA is complete (cRORA) when this value is higher and corresponds to geographic atrophy (GA).

### 2.4. Statistical Analysis

We analyzed both right and left eyes. In this study, we considered only the results from the right eyes for statistical analysis in order not to overestimate the statistical significance and *p*-values. Analysis from right and left eyes showed substantially similar results.

The data were analyzed by parametric or non-parametric analyses, depending on their distribution. Both Pearson’s correlation and Spearman rank order correlation were used. ERG data were log transformed to better approximate normal distribution. A *p* value of less than 0.05 was considered statistically significant.

**Table 1 diagnostics-13-00850-t001:** Molecular Genetic Data of EYS patients.

ID	Sex	Nucleotide Change	Amino Acid Change	Allele State	Varsome	ACMG Criteria				dbSNP rs	References
1	M	c.8411_8412insTT	p.(Thr2805*)	HOM	LP	PVS1	PM2			NA	[11]
2	F	c.8598del	p.(Gly2867Valfs*5)	HOM	P	PVS1	PM2	PP5		rs1050742628	NA
3	M	c.8161_8165del	p.(Gln2721Alafs*24)	HET	LP	PVS1	PM2			NA	NA
		c.9405T>A	p.(Tyr3135*)	HET	P	PVS1	PM2	PP5		rs137853190	[6]
4	M	c.5928-2A>G		HOM	P	PVS1	PM2	PP5		rs181169439	[2]
5	M	c.5621dup	p.(Pro1875Thrfs*8)	HET	LP	PVS1	PM2			NA	[22]
		c.8411_8412insTT	p.(Thr2805*)	HET	LP	PVS1	PM2			NA	[11]
6	M	c.8565_8568del	p.(Asn2855Lysfs*5)	HET	P	PVS1	PM2	PP5		rs1216993077	NA
		c.4073del	p.(Pro1358Glnfs*23)	HET	LP	PVS1	PM2			NA	NA
7	F	c.5644+5G>T		HOM	LP	PM2	PP3			NA	NA
8	F	c.4045C>T	p.(Arg1349*)	HET	P	PVS1	PM2	PP5		rs930421180	[2]
		c.4350_4356del	p.(Ile1451Profs*3)	HET		PVS1	PM2	PP5		rs761238771	[2]
9	F	c.7919G>A	p.(Trp2640*)	HOM	P	PVS1	PM2	PP5		rs527236066	[2]
10	F	c.403_423delinsCTTTT	p.(Thr135Leufs*26)	HET	P	PVS1	PM2	PP5		rs1582376398	[23]
		c.(2135_2204)_(2351_2469)del		HET	LP	PVS1	PM2			NA	NA
11	F	c.(2137+1_2138-1)_(2259+1_2260-1)del		HOM	LP	PVS1	PM2			NA	[24]
12	M	c.4045C>T	p.(Arg1349*)	HET	P	PVS1	PM2	PP5		rs930421180	[25]
		c.9299_9302del	p.(Thr3100Lysfs*26)	HET	P	PVS1	PM2	PP5		rs769824975	[26]
13	F	c.9328G>A	p.(Gly3110Ser)	HOM	VUS	PM2	PM5	PP3		NA	NA
14		c.5621dup	p.(Pro1875Thrfs*8)	HET	LP	PVS1	PM2			NA	[22]
		c.8411_8412insTT	p.(Thr2805*)	HET	LP	PVS1	PM2			NA	[11]
15	M	c.5928-2A>G		HOM	P	PVS1	PM2	PP5		rs181169439	[2]
16	F	c.4219C>T	p.(Gln1407*)	HET	LP	PVS1	PM2			rs1421392730	NA
		del ex32-35		HET	LP	PVS1	PM2			NA	NA
17	M	*c.1852G>A*	*p.(Gly618Ser)*	HET	VUS	PM2	PP5	BP4		rs142450703	[2]
		*c.1561_1563del*	*p.(Asn521del)*	HET	VUS	PM2	PM4	PP5		rs747069281	NA
		c.2309A>C	p.(Gln770Pro)	HET	VUS	PM2	BP4			rs398123574	[27]

Legend: F, female; M, male; HET, heterozygous; HOM, homozygous; in *italics*, *in cis* variants; NA, not available; VUS, variant of unknown significance; LP, likely pathogenic; P, pathogenic.

## 3. Results

Clinical results of individual patients are reported in Table 2. The results of RP-SSS, ERGs and iRORA area are reported in Appendix A.

Figure 1 shows typical macular appearance in an EYS patient with a mild (Pt #3) and an advanced RP-SSS (Pt #1) according to the staging classification by Iftkhar [9]. In the advanced patient EZ is not detectable and ONL thickness is markedly reduced in comparison to the patient with an early stage of the disease. Figure 1 shows the SRI analysis in the macula obtained from the moderate (A) and the advanced (B) EYS patient. SRI area increases with disease severity.

Patient A has a RP-SSS of 1 and 12 corresponding to grade 1 and 4, respectively.

This picture is associated with changes in the sub-RPE slab (areas of increased SRI) within the 5 mm circle outlined in white. The red line shows the atrophic area closest to the fovea.

The staging score was significantly correlated with age and disease duration (r = 0.54, *p* < 0.01) as shown in Figure 2A,B. An age of 45 years and a disease duration of 15 years corresponded to the appearance of an advanced RP score (≥8). The data showed a trend to saturate after the age of 50, indicating a ceiling effect.

Figure 3 shows a scattergram of SRI area as a function of staging score. It can be noted that SRI area, and consequently the CRA severity, was positively correlated with severity score (r = 0.5, *p* < 0.05).

SRI did not show any significant correlation with 30 Hz Flicker or focal ERG amplitude.

Figure 4A,B shows scattergrams of SRI area as a function of LogMAR acuity and EZ extension. SRI area was significantly correlated with both measurements (r = 0.5, *p* < 0.01).

## 4. Discussion

The present study was designed to evaluate the severity stage of EYS disease patients, as determined by the RP-SSS [9], and to estimate the area of CRA (from a standardized method measuring the area of SRI in the macula) in relation to disease stage in the same patients. The results of our study showed a correlation between the stage of RP and the age and disease duration of patients. Such correlation revealed a severe stage at a relatively early age of 45, supporting the severity of the molecular pathology underlying this sub-type of RP.

Previous scientific investigations concerning structure and function correlations in patients with biallelic mutations in the EYS gene support a rapid disease progression as a function of time. McGuigan et al. [11] evaluated a cohort of 15 patients by chromatic static perimetry, SD-OCT, and enface autofluorescence imaging reporting anomalies in the outer nuclear retinal layer of the central retina with some exceptions in the foveal region. Furthermore, an analysis of the perifoveal region as a function of time showed that photoreceptor structural loss was followed by dysmorphology of the inner retina and loss of retinal pigment epithelial integrity. Based on their results, arRP caused by EYS mutation was considered a more rapidly progressive disease compared to other gene mutations causing arRP, such as the USH2A and MAK variants. Some authors suggested that a different phenotype and progression in EYS-related retinal degeneration could be related to the specific EYS genotype [26,28,29,30,31]. However, due to the limited number of patients examined, such correlations could not be verified and confirmed in the present study.

An important finding of this study is represented by the correlation of CRA area with RP-SSS. The increase in CRA area was positively correlated with a greater RP-SSS value, indicating an increase in the disease severity. In addition, it was correlated inversely with the EZ width and positively with LogMAR acuity. Although other morphological predictor factors of atrophy progression in OCT analysis have been reported [32], RORA is the result of multiple information on the status of EZ, outer segments of the photoreceptors, interdigitation zone and RPE-Bruch’s membrane complex [33,34,35]. Recently, RORA assessment has been considered not only for early diagnosis but also as a prognostic factor of AMD progression [21]. Indeed, the possibility of evaluating the progression of central atrophy growth could become a key element in the evolutionary study of RP. Currently, the detection of RORA finds an important application in eyes with age-related macular degeneration (AMD) [36]. Automated identification of RORA using machine learning has also been described [37]. Extending the present approach to inherited retinal diseases would be important for predicting their natural history. Indeed, we already demonstrated in other studies how the SRI area, as a result of RORA, was correlated with the disease severity stage in patients with USH2A-related Retinitis Pigmentosa [10].

Although EYS-related RP represents a rare disease, our sample did not include a high number of patients. This constitutes a limitation to our study. Further comparative studies using a similar methodology would be desirable in order to identify useful diagnostic and prognostic criteria to study the natural history of the EYS-related disease, which is currently still difficult to predict.

In our study population, no correlations were found between the stage of the disease and the ERG, both focal and 30 Hz flicker ERG. Although in most cases the degenerative evolution of the disease proceeds from rod to cone photoreceptors and the cone loss is considered a secondary occurrence, the obtained ERG data show that a severe cone dysfunction may be present early in EYS patients. Similar to what is observed in patients with USH2A mutations [20], our study patients showed severely abnormal 30 Hz flicker ERG and fERG, well in advance of other cone-related visual tests, such as visual acuity. This finding supports an early involvement of cone driven responses in EYS-related RP. This hypothesis is in agreement with McGuigan et al. [11] who reported the constant presence of anomalies in the outer nuclear retinal layer of the central retina in a cohort of patients with EYS mutations, except in some where only the foveal region appeared still preserved. It is important to specify that, although the number of patients who underwent a FERG in this study was 7 out of 15, all patients had abnormal ERG responses. Additionally, 30 Hz Flicker ERG showed markedly reduced responses in all patients and was unrelated to other parameters. This latter finding indicates that in our patients a severe extrafoveal cone dysfunction was present, too.

Although no treatments are available for EYS-related RP, many novel therapeutic approaches could represent potential therapeutic options in the future [38]. In view of the above, the correct identification of useful outcome measures to evaluate the effectiveness of potential therapies in upcoming clinical trials is a goal to be achieved. Studies in this regard are still few and more will have to be conducted to strengthen our hypothesis.

The data reported in this study qualifies the SRI in the detection of iRORA as an important parameter concerning the rate of progression of retinal degeneration caused by biallelic mutations in the EYS gene.

## 5. Conclusions

This research has provided new information concerning patients affected by Retinitis Pigmentosa associated with EYS gene mutations. We used a known RP severity score and validated techniques for the detection of morpho-functional correlations. In particular, in EYS-related RP the severity score was positively correlated with the central retinal atrophy, the central visual acuity and the EZ extension and increases with age and disease duration. In contrast, due to the intrinsic characteristics of the disease, functional measures did not appear sensitive enough to reveal staging-based changes. Given the small size of our sample, further studies concerning the natural history of this kind of retinal dystrophy would be desirable.

## Figures and Tables

**Figure 1 diagnostics-13-00850-f001:**
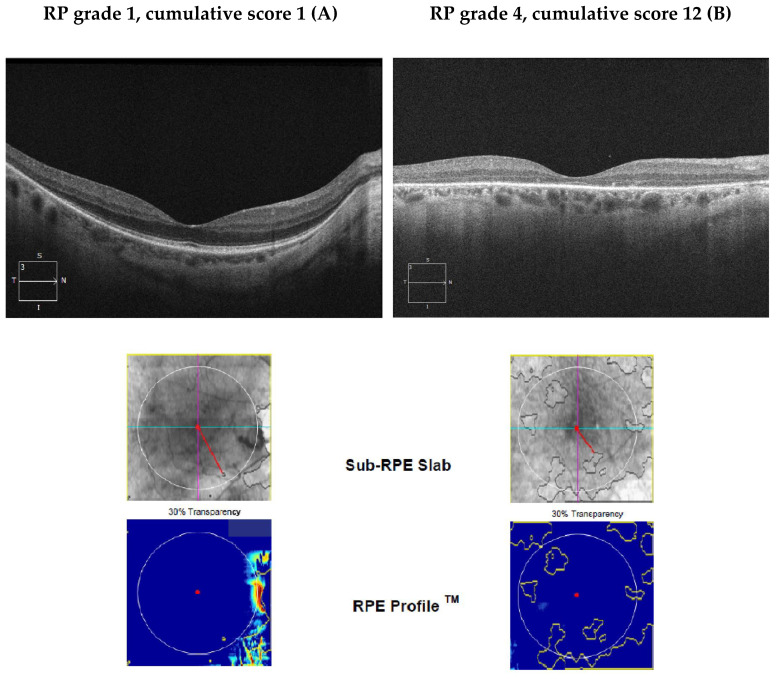
Images OCT B scan, sub-RPE slab and RPE profile of two patients with early stage RP (**A**) and advanced stage RP (**B**). In early stage RP (**A**), the ellipsoid zone is preserved in the foveal and parafoveal regions. In advanced RP (**B**) the almost complete loss of the ellipsoid zone and marked reduction in the outer nuclear layer are observable.

**Figure 2 diagnostics-13-00850-f002:**
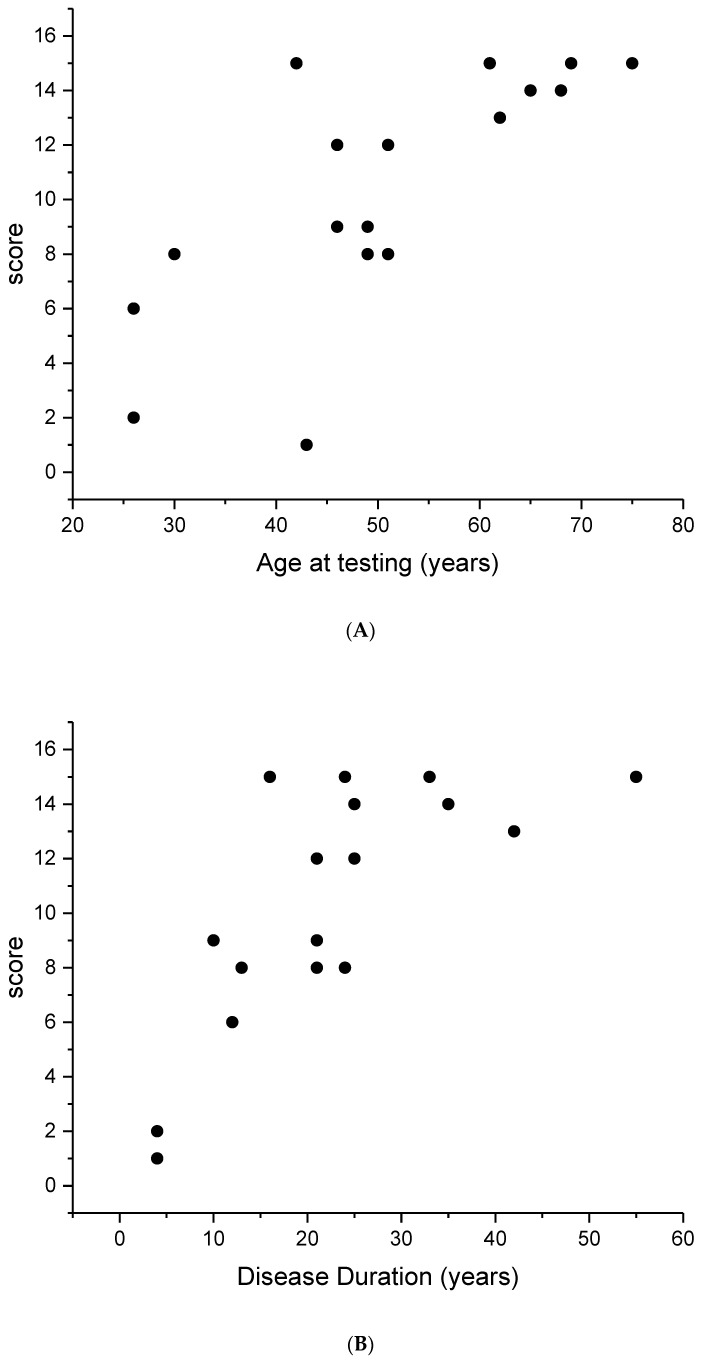
Cumulative score (CS) from the right eye of each EYS patient plotted as a function of age at testing (**A**) and disease duration (**B**). It can be noted that the score increases linearly with both parameters. The r value is 0.54 (*p* < 0.01).

**Figure 3 diagnostics-13-00850-f003:**
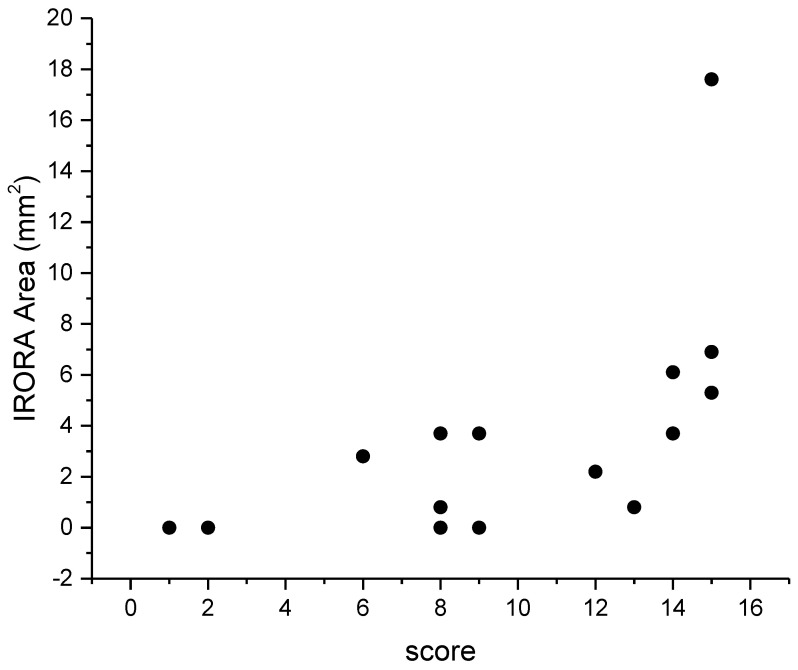
Results of correlation analysis between SRI and CS. SRI showed a positive correlation with CS. The r value is 0.5 (*p* < 0.05).

**Figure 4 diagnostics-13-00850-f004:**
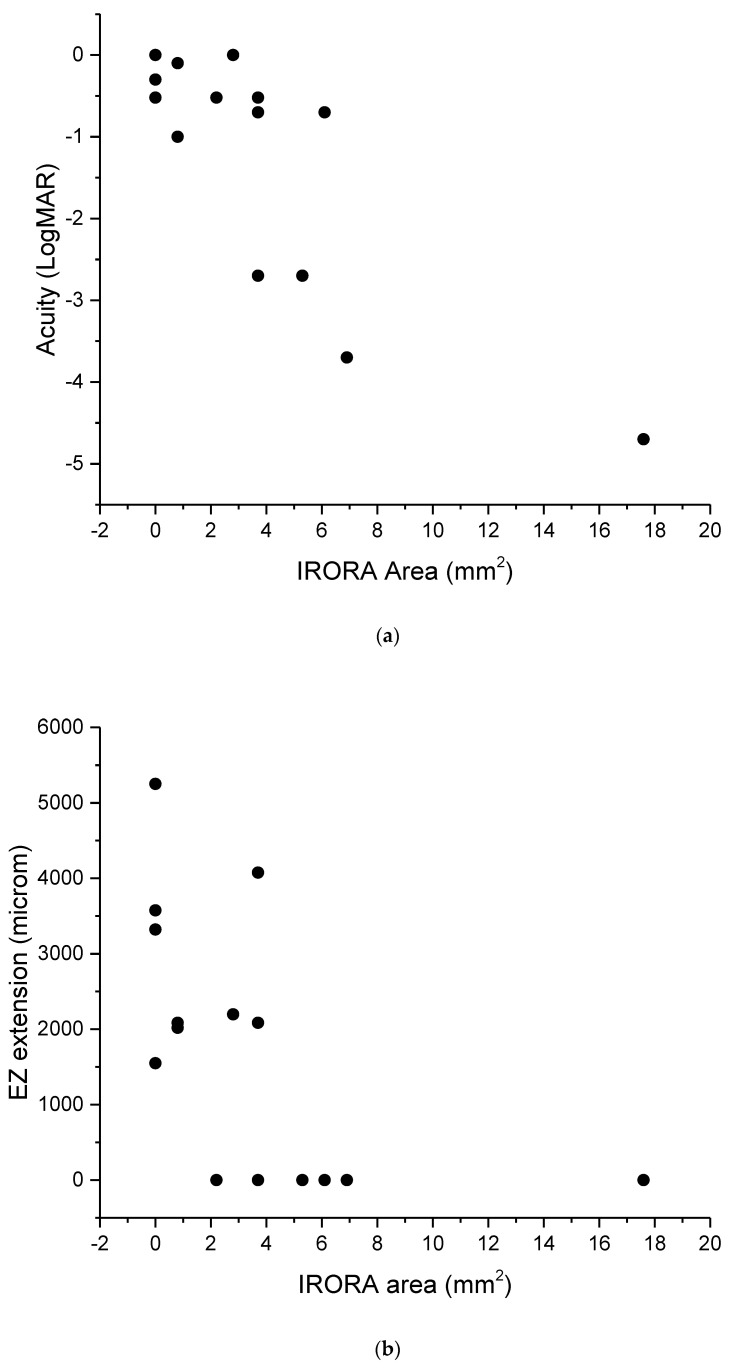
Correlation analysis between SRI and BCVA (**a**) and EZ extension (**b**). SRI area gradually increases as visual acuity and/or ellipsoid zone extension decreases.

**Table 2 diagnostics-13-00850-t002:** Demographic and clinical data of studied patients.

Nr	Sex	Onset	Age of Assessment	Disease Duration	RE			LE		
					BCVA (LogMAR)	VF	EZ	BCVA (LogMAR)	VF	EZ
1	M	25	46	21	−0.50	21	0	−1.0	25	0
2	F	42	75	33	−2.70	0	0	−0.50	10	0
3	M	39	43	4	0.00	124	5251	−0.10	131	5053
4	M	18	42	24	−3.70	0	0	−3.70	0	0
5	M	17	30	13	−0.50	50	3321	−0.18	43	641
6	M	22	26	4	0.00	129	3576	0.00	133	3525
7	F	20	62	42	−0.92	17	2084	−0.80	18	1337
8	F	14	26	12	0.00	79	2197	0.00	80	2577
9	F	25	49	24	−0.70	120	2085	−0.50	107	1556
10	F	25	46	21	−0.30	62	1549	−0.30	60	1581
11	F	43	68	25	−2.70	17	0	−2.70	0	0
12	M	39	49	10	−0.50	30	4075	−0.40	16	4313
13	F	30	65	35	−0.70	18	0	−0.50	16	0
14	F	30	51	21	−0.10	22	2018	0.00	25	2448
15	M	14	69	55	−3.70	0	0	−3.70	0	N/A
16	F	26	51	25	−0.30	19	0	−0.18	20	3395
17	M	45	61	16	−2.70	0	0	−2.70	0	0

Legend: F, female; M, male; RE, right eye; LE, left eye; BCVA, best corrected visual acuity; VF, visual field; EZ, ellipsoid zone.

## Data Availability

Data available from authors.

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
