# Peer review of "Retinitis Pigmentosa Associated with EYS Gene Mutations: Disease Severity Staging and Central Retina Atrophy"

_diagnostics, 2023, doi:10.3390/diagnostics13050850_

Round 1

Reviewer 1 Report

The aim of the manuscript entitled Retinitis Pigmentosa associated with EYS gene mutations: Disease Severity Staging and Central Retina Atrophy” by Giorgio Placidi et al. is to determine the disease severity stage in a cohort of ArRP patients carrying pathogenic variants in the EYS gene and to estimate in these patients the extent of central retina atrophy (CRA) by using the automated SRI measurements.  17 subjects affected by RP due to variants in the EYS gene were studied and the observations and results are discussed.

The paper is generally well written.  The manuscript is correct in terms of the structure and the merits. The Abstract, the Introduction and the Discussion sections are clear and provide all the needed information for the readers. The Material and Methods are clearly described, the statistical analysis is performed appropriately.  The manuscript appears to be appropriate for Diagnostics journal  and may be of potential interest to specialists.

I have a few minor concerns, that should be addressed.

Minor concerns:

1.The main disadvantage of the manuscript is the small sample of studied patients (despite it is a rare disease) and in my opinion it should be highlighted as a limitation of the study.

2.Define abbreviations cRORA and GA  in the text in the first mention (line 153)  

3. Concerning the Results section, I would suggest some little changes:

Table 3 is too extensive and difficult to follow, some data in table 3 overlap with table 2 (sex, onset, age of assessment, disease duration). Consider including table 3 as an additional file that is not in the main text.

Author Response

Review Report 1

The aim of the manuscript entitled “Retinitis Pigmentosa associated with EYS gene mutations: Disease Severity Staging and Central Retina Atrophy” by Giorgio Placidi et al. is to determine the disease severity stage in a cohort of ArRP patients carrying pathogenic variants in the EYS gene and to estimate in these patients the extent of central retina atrophy (CRA) by using the automated SRI measurements.  17 subjects affected by RP due to variants in the EYS gene were studied and the observations and results are discussed.

The paper is generally well written.  The manuscript is correct in terms of the structure and the merits. The Abstract, the Introduction and the Discussion sections are clear and provide all the needed information for the readers. The Material and Methods are clearly described, the statistical analysis is performed appropriately.  The manuscript appears to be appropriate for Diagnostics journal  and may be of potential interest to specialists.

I have a few minor concerns, that should be addressed.

Minor concerns:
Point 1. The main disadvantage of the manuscript is the small sample of studied patients (despite it is a rare disease) and in my opinion it should be highlighted as a limitation of the study.

Response 1: A sentence concerning this issue has been added in the “Discussion” section (see lines 275-279).

Point 2. Define abbreviations cRORA and GA  in the text in the first mention (line 153)  

Response 2: Done (see lines 151-152 and abbreviations)

Point 3. Concerning the Results section, I would suggest some little changes:
Table 3 is too extensive and difficult to follow, some data in table 3 overlap with table 2 (sex, onset, age of assessment, disease duration). Consider including table 3 as an additional file that is not in the main text.

Response 3: Data concerning sex, onset, age of assessment and disease duration in Table 3 have been deleted. This table has been moved to “Supplementary Material” as suggested.

We hope that changes we made are in the right direction to improve the quality of the manuscript

Best regards
Dr Giorgio Placidi

Reviewer 2 Report

This work is about retinitis pigmentosa associated with eye shut homolog gene mutations analyzing disease severity staging and central retina atrophy in patients with autosomal recessive retinitis pigmentosa. It is an interesting article about to diagnostic of the pathology but I have some point to be addressed before it´s acceptation:

Abstract has a reference number. Please delete it from the text. This section and conclusion section as well have no references at all in a scientific journal.

A cohort of EYS patients was studied. What about a control cohort?? Wouldn't it be much more reliable to determine the effectiveness of the diagnostic system than to compare it with a control sample? It would be convenient to know from the morphological point of view how the macula is in control patients and thus determine exactly the deviation in cases patients with autosomal recessive retinitis pigmentosa, if what is wanted is to demonstrate the effectiveness of the technique for its detection. Please explain it.

I would like to know why the left eyes have been removed to do the statistical tests since the n would be much higher if they were included. In any case, the paper does not explain it and should be explained in materials and methods.

Conclusion has a reference number as well. Please removed it from the text.

This section state "The data reported in this study qualify the SRI in the detection of iRORA as 296 an important biomarker concerning the rate of progression of retinal degeneration caused 297 by biallelic mutations in the EYS gene". I think the conclusions are not conclusive. As far as I understand, here in this study, a biomarker has not been developed but a system for detecting the morphology of the affected area. The rest of the conclusions are wishes, not conclusions, that should be at the end of the discussion. This section should include the conclusions of the results about the disease severity staging and central retina atrophy obtained in the work not the hope of its subsequent use. Please redo this section with true conclusions from the work done.

Author Response

Review Report 2

This work is about retinitis pigmentosa associated with eye shut homolog gene mutations analyzing disease severity staging and central retina atrophy in patients with autosomal recessive retinitis pigmentosa. It is an interesting article about to diagnostic of the pathology but I have some point to be addressed before it´s acceptation:

Point 1. Abstract has a reference number. Please delete it from the text. This section and conclusion section as well have no references at all in a scientific journal.

Response 1. Done.

Point 2. A cohort of EYS patients was studied. What about a control cohort?? Wouldn't it be much more reliable to determine the effectiveness of the diagnostic system than to compare it with a control sample? It would be convenient to know from the morphological point of view how the macula is in control patients and thus determine exactly the deviation in cases patients with autosomal recessive retinitis pigmentosa, if what is wanted is to demonstrate the effectiveness of the technique for its detection. Please explain it.

Response 2. The main outcome of this research is represented by the correlation between the disease severity stage and the extent of CRA in a cohort of EYS-related RP patients by using the automated SRI measurements.

Since in healthy subjects the subRPE illumination is not detectable due to the non-permeability of the RPE, results obtained from a control group were not reported.

However, a control cohort of 10 normal subjects was considered. In all subjects there were no areas of retinal atrophy. Hence, the quantitative value of the subRPE illumination always corresponded to 0.

Point 3. I would like to know why the left eyes have been removed to do the statistical tests since the n would be much higher if they were included. In any case, the paper does not explain it and should be explained in materials and methods.

Response 3. In this study, We considered only the results from one eye in order to not overestimate the p-values and following a criterion already used in another work (Falsini, B.; Placidi, G.; De Siena, E.; Savastano, M.C.; Minnella, A.M.; Maceroni, M.; Midena, G.; Ziccardi, L.; Parisi, V.; Bertelli, M.; et al. USH2A-Related Retinitis Pigmentosa: Staging of Disease Severity and Morpho-Functional Studies. Diagnostics 2021, 11, 213, doi:10.3390/diagnostics11020213). In our overall analysis, We also examined both eyes and the results were basically similar. All the data, included those related to the LE are reported in Tables 2 and 3.

A clarifying sentence has been added at the beginning of the “2.4. Statistical analysis” paragraph (lines 155-158).

Point 4. Conclusion has a reference number as well. Please removed it from the text.

Response 4. Done.

Point 5. This section state "The data reported in this study qualify the SRI in the detection of iRORA as an important biomarker concerning the rate of progression of retinal degeneration caused by biallelic mutations in the EYS gene". I think the conclusions are not conclusive. As far as I understand, here in this study, a biomarker has not been developed but a system for detecting the morphology of the affected area. The rest of the conclusions are wishes, not conclusions, that should be at the end of the discussion. This section should include the conclusions of the results about the disease severity staging and central retina atrophy obtained in the work not the hope of its subsequent use. Please redo this section with true conclusions from the work done.

Response 5. The sentence containing reference #38 has been moved to the end of the "Discussion" section (lines 300-308). We changed the term “biomarker” with the term “parameter” (see line 307) and more appropriate conclusions have been developed as recommended (see lines 311-319).

We hope that changes we made are in the right direction to improve the quality of the manuscript

Best regards

Dr Giorgio Placidi

Round 2

Reviewer 2 Report

Thank you for the modifications.